# Characterization of the First Animal Toxin Acting as an Antagonist on AT1 Receptor

**DOI:** 10.3390/ijms24032330

**Published:** 2023-01-24

**Authors:** Anne-Cécile Van Baelen, Xavier Iturrioz, Marion Chaigneau, Pascal Kessler, Catherine Llorens-Cortes, Denis Servent, Nicolas Gilles, Philippe Robin

**Affiliations:** Département Médicaments et Technologies pour la Santé (DMTS), Université Paris-Saclay, F-91191 Gif-sur-Yvette, France

**Keywords:** conotoxin, angiotensin, AT1, GPCR, *Conus miliaris*

## Abstract

The renin-angiotensin system (RAS) is one of the main regulatory systems of cardiovascular homeostasis. It is mainly composed of angiotensin-converting enzyme (ACE) and angiotensin II receptors AT1 and AT2. ACE and AT1 are targets of choice for the treatment of hypertension, whereas the AT2 receptor is still not exploited due to the lack of knowledge of its physiological properties. Peptide toxins from venoms display multiple biological functions associated with varied chemical and structural properties. If Brazilian viper toxins have been described to inhibit ACE, no animal toxin is known to act on AT1/AT2 receptors. We screened a library of toxins on angiotensin II receptors with a radioligand competition binding assay. Functional characterization of the selected toxin was conducted by measuring second messenger production, G-protein activation and β-arrestin 2 recruitment using bioluminescence resonance energy transfer (BRET) based biosensors. We identified one original toxin, A-CTX-cMila, which is a 7-residues cyclic peptide from *Conus miliaris* with no homology sequence with known angiotensin peptides nor identified toxins, displaying a 100-fold selectivity for AT1 over AT2. This toxin shows a competitive antagonism mode of action on AT1, blocking Gαq, Gαi3, GαoA, β-arrestin 2 pathways and ERK_1/2_ activation. These results describe the first animal toxin active on angiotensin II receptors.

## 1. Introduction

Venoms constitute a vast library of biochemically stable peptide toxins with particular pharmacological properties, which have evolved to provide their host with capture or defense capabilities. It is estimated that the 200,000 species of venomous animals existing on earth could produce around 40 million toxins, whose properties are still largely unexploited [1]. Toxins are peptides often containing cysteines, giving them different cross-linking patterns and conferring various three-dimensional functional structures. They are also peptides with high selectivity and specificity for their target. In addition, the biological effects and molecular targets of many toxins present in venoms are still unknown, which explains why venom screening to identify new ligand-receptor pairs has gained momentum. Toxins have already proven their interest in the development of therapeutics [2]. For instance, one peptide derived from *Conus magus* (Ziconotide) has received FDA approval as an analgesic targeting Ca_V_2.2 [3]. So far, only 7000 toxins have been fully characterized, mostly on ionic channels, but very few of them act on GPCR which are nevertheless the therapeutic targets of 30% of drugs currently used. GPCR-interacting toxins can be classified into 2 groups: agonist-mimicking peptides, sharing sequence and structural similarities with endogenous ligands and primarily activating receptors; and others that are structurally different from the endogenous ligands, and exhibit antagonist, allosteric, or agonist modes of action [4].

The RAS is involved in the regulation of numerous biological functions, particularly in the regulation of blood pressure [5,6]. The main axis is composed of angiotensinogen, which is converted by renin into angiotensin I and then into angiotensin II (AngII) by the angiotensin-converting enzyme (ACE) [7,8]. AngII, and its downstream metabolite AngIII, are the main effectors of the system and target the GPCRs angiotensin type 1 receptor (AT1) and angiotensin type 2 receptor (AT2) [9,10]. Once activated, the AT1 receptor induces several physiological effects such as vasoconstriction, increase in blood pressure, hypertrophy of cardiomyocytes and cardiac remodeling [11,12]. In addition to its actions at the cardiovascular level, AT1 activation is associated with profibrotic [13], pro-inflammatory [14] and pro-apoptotic [15] effects. Activated AT1 receptor is known to couple with G-proteins of the Gαq and Gαi/o families [16]. Upon activation, this receptor is also phosphorylated by GPCR kinases (GRK), enabling β-arrestin to bind to the receptor, terminating G protein-mediated signaling and leading to receptor internalization. This β-arrestin recruitment activates specific pathways in a G-protein-independent manner, such as ERK_1/2_ activation [17,18].

The AngII/AT2 axis generally has opposite effects to the AngII/AT1 axis and is described as having hypotensive and vasodilator actions [19,20]. AT2 receptors also have antifibrotic, anti-inflammatory, neuroprotective, and antiapoptotic effects and increase natriuresis [21,22]. Recently, its anti-inflammatory action has been highlighted, showing an interest in the treatment of other pathologies such as gout or arthritis in the mouse model [23,24]. AT2 receptor activation may promote eNOS activation [25]. However, even if the AT2 receptor is classified as a GPCR and possesses all the GPCR common features, its effective coupling to classical G-protein-dependent pathways is still a matter of debate. While some studies show its coupling to Gαi proteins [26,27], others show the opposite [28]. The AT2 receptor also does not interact with β-arrestins and is not internalized after stimulation by AngII [29]. These unconventional properties of the AT2 receptor could be due to its structure itself and, in particular, to the behavior of its C-terminal helix VIII positioned in such a way that it blocks G-protein/β-arrestin binding sites [30]. Its functional study is still not fully explored due to the difficulty of highlighting any signaling pathway.

The RAS is already the target of one toxin-derived anti-hypertensive drug, namely, captopril. This toxin, extracted from the venom of *Bothrops jararaca*, is an inhibitor of ACE leading to a decrease in AngII production [31]. The RAS is also the target of small organic molecules, the AT1 blockers ‘sartans’. However, 30% of patients under anti-hypertensive drug treatment are non-responders, either due to poor observance due to side effects or due to ineffectiveness of the treatment [32]. Indeed, even today, the signaling pathways involving AT1 and AT2 receptors are not yet fully described, in particular, because of the lack of pharmacological tools allowing their study. The aim of the present study is therefore to find new tools to allow the functional study of these receptors. As mentioned before, toxins might fulfill this role. Today there are very few identified toxin-GPCR couples despite the huge implication of GPCRs in the therapeutic field. Despite the existence of toxin-derived peptide inhibiting ACE, there are still no toxin identified targeting angiotensin receptors and only very few ligands selective for AT2 receptors are available [4]. Here we describe the identification and the pharmacological characterization of the first toxin acting on AngII receptors.

## 2. Results

### 2.1. Screening on Angiotensin II Receptors

With the aim of identifying toxins targeting the AT2 receptor, part of the Venomics toxin library was screened on human AT2 receptor by competition binding assays with ^125^I-Sar1-Ile8-AngII radioligand [1]. Among obtained hits, one toxin exhibited no sequence homology neither with other known toxins nor with angiotensin peptides. This toxin, named A-CTX-cMila, with standing for activity on angiotensin II receptors [33], is a 7-residue cyclic peptide (CHFWVCP) whose sequence was obtained by an omics analysis of *Conus miliaris*. A-CTX-cMila is a small conotoxin containing only one disulfide bridge between Cys1 and Cys6. This cysteine pattern is one of the lesser represented for conotoxins, which often contain more than two cysteine residues and more than one disulfide bridge [34].

### 2.2. Chemical Production of the Toxin

To perform the complete functional characterization of A-CTX-cMila, the toxin was produced by solid-phase peptide synthesis (SPPS) using Fmoc-based strategy [35] on a peptide synthesizer. The reduced peptide was purified by preparative HPLC prior to oxidation, with an optimized protocol to form the disulfide bridge and obtain the final functional peptide. A last step of preparative HPLC purification of the oxidized A-CTX-cMila is necessary to obtain the final product. From an initial quantity of loaded resin of 25 µmoles, 3.3 mg of final toxin is obtained, i.e., 15% yield compared to the expected quantity of loaded resin. The final form of the peptide is surprisingly more hydrophobic with a shift of acetonitrile of 3% higher than the reduced form (Figure 1A). The analysis of the final product by LC-ESI-MS shows a unique mass at m/z 889.3 (Figure 1B) consistent with its calculated theoretical m/z of 889.3489 for oxidized A-CTX-cMila.

### 2.3. Binding Properties of A-CTX-cMila on Angiotensin II Receptors

The binding properties of A-CTX-cMila to human AT2 receptor were determined by radioligand competition binding assay. The binding curve of ^125^I-Sar1-Ile8-AngII (^125^I-S1I8-AngII) in the presence of increasing concentrations of A-CTX-cMila shows that the toxin inhibits the binding of the radioligand in a dose-dependent manner and binds to AT2 receptor with a K_i_ of 27.5 ± 2.1 µM (*n* = 5) (Figure 2A). AT1 and AT2 receptors belonging to the same phylogenetic branch and sharing the same endogenous ligands (AngII and AngIII) and the binding of the toxin on the AT1 receptor was investigated. As shown in Figure 2A, the toxin is also able to inhibit the binding of ^125^I-Sar1-Ile8-AngII on the AT1 receptor and exhibits 100-fold higher affinity for this receptor (K_i_ of 0.32 ± 0.13 µM, *n* = 5). To determine whether the toxin binds on living cells and the extracellular side, the effect of A-CTX-cMila on AngII binding was then tested on intact living cells expressing AT1 or AT2 receptors. For this purpose, A-CTX-cMila effect on the binding of fluorescent Cy5-AngII to AT1- or AT2-expressing CHO cells was evaluated by flow cytometry. The results (Figure 2B) show that the non-specific labelling of cells measured in the presence of 1 µM non-labelled AngII and 20 nM Cy5-AngII is very low (7% ± 2%) compared to the total labelling observed with Cy5-AngII alone, for both receptors. In the presence of 100 µM A-CTX-cMila, the labelling of cells was also strongly reduced (74% ± 7% inhibition of the specific labelling for AT1 and 75% ± 9% for AT2) indicating that the toxin is able to inhibit Cy5-AngII binding on AT1 and AT2 receptors. To confirm the flow cytometry results, the labelling of Cy3-AngII to CHO cells expressing AT1 or AT2 was evaluated qualitatively by fluorescence microscopy. The results show that after incubation for 30 min at 37 °C, Cy3-AngII staining of AT1-expressing cells appears as small dots distributed throughout the cells with no apparent membrane labeling, suggesting the internalization of Cy3-AngII in endocytosis compartments. This result is fully consistent with the well-established property of AT1 to be rapidly internalized in the cell after AngII binding. The staining of AT2-expressing cells appears diffuse and more concentrated on the cell contour consistent with a cell surface localization of the receptor, which is described as not undergoing internalization upon AngII binding. In the same conditions, the addition of the toxin before Cy3-AngII, leads to an almost complete loss of the staining, meaning that the toxin inhibits the binding of Cy3-AngII on both cell types and thus prevents its internalization into the cells. These observations suggest that the toxin binding site is localized on the extracellular side of the AT1 and AT2 receptors (Figure 2C).

### 2.4. Pharmacological Characterization of A-CTX-cMila on AT1-Mediated Gαq/PLC Pathway

A-CTX-cMila binds to the AT2 receptor with a poor affinity compared to the AT1 receptor. Considering both the much better affinity of the toxin for the AT1 receptor, and that the functional characterization of the AT2 receptor is still laborious on heterologous systems, pharmacological characterization was conducted on the AT1 receptor.

The AT1 receptor interacts with heterotrimeric G-proteins, and especially Gαq proteins. This activation leads to intracellular calcium flux and the activation of protein kinases [11]. The impact of A-CTX-cMila on the Gαq/Phospholipase C (PLC) pathway after binding on the AT1 receptor was assessed by measuring the production of inositol phosphates. In this aim, inositol monophosphate (IP1), the dephosphorylated product of inositol trisphosphate (IP3) that accumulates in the cell in the presence of the IP1 phosphatase inhibitor LiCl, was quantified. IP1 production is measured by homogeneous time-resolved fluorescence (HTRF) assay kits.

As expected, AngII induces IP1 production in a dose-dependent manner with an EC_50_ of 6.6 ± 1.1 nM (*n* = 6) (Figure 3A) and a maximal effect of 16-fold the basal level. Under the same experimental conditions, the toxin does not induce any agonist effect on this pathway (Figure 3A). In the presence of 10 nM AngII, the toxin causes the inhibition of IP1 production in a dose-dependent manner, with an IC_50_ of 23.7 ± 4.6 µM (*n* = 3) (Figure 3B). To better characterize the antagonist mode of action, dose-response curves of AngII were performed in the presence of various concentrations of the toxin ranging from 20 to 300 µM. Results demonstrate that when the doses of toxin increase, the EC_50_ of AngII raises from 6.6 ± 1.1 nM to 39.0 ± 6.7 nM with 20 µM toxin, to 71.9 ± 8.1 nM with 40 µM toxin, to 154 ± 20 nM with 75 µM toxin, to 272 ± 31 nM with 150 µM toxin, to 359 ± 39 nM with 300 µM toxin (*n* = 3), without any significant change in the maximal response (Figure 3C). Schild’s representation of these values is linear with a slope of −1.2 ± 0.2 and a pA2 of 4.9 ± 0.3 (*n* = 3) (Figure 3D). The associated K_B_ value is 17.1 ± 6.9 µM (*n* = 3). In conclusion, A-CTX-cMila behaves as a competitive antagonist on the Gαq/PLC pathway.

### 2.5. Pharmacological Characterization of A-CTX-cMila on AT1-Mediated Gαi3 and GαoA Pathways

AT1 receptor is also known to recruit other G-proteins such as Gαi3 and GαoA, leading to an increase in intracellular calcium release. The impact of the toxin on the activation of these G-proteins was measured using specific BRET biosensors developed by M. Maziarz et al. [36].

The pharmacological profile of A-CTX-cMila was determined on the Gαi3 pathway. If the receptor activates the G-protein, the trimer Gαβγ dissociates, and the KB-1753-Nanoluc is able to recognize specifically the GTP-loaded Gα form linked to a yellow fluorescent protein (YFP), leading to an increase of the BRET signal. 10 nM AngII increased the BRET signal, reflecting the activation of Gαi3 protein by AngII. The kinetic study shows that this activation reaches a peak one minute after the AngII addition (Figure 4A,B). Neither candesartan, an AT1 receptor antagonist, nor the toxin alone could induce the activation of Gαi3 (Figure 4A). The activation is fully inhibited by 100 µM of A-CTX-cMila as well as 1 µM of candesartan (Figure 4B). Thus, the toxin has no agonist mode of action but an antagonist mode of action on this pathway. In the presence of 50 nM AngII, the toxin inhibits the AngII-induced BRET signal in a dose-dependent manner with a micromolar potency (IC_50_ = 28.5 ± 12.0 µM, *n* = 3) (Figure 4C). AngII induces the Gαi3 activation in a dose-dependent manner with an EC_50_ of 4.9 ± 0.9 nM (*n* = 3) (Figure 4D). In presence of 150 µM A-CTX-cMila, the AngII dose-response curve is shifted to the right, raising the EC_50_ of AngII to 468 ± 219 nM without affecting the maximal effect of AngII (*n* = 3) (Figure 4D), the sign of a competitive antagonist mode of action of A-CTX-cMila on Gαi3 activation.

The effect of A-CTX-cMila was then tested on the GαoA pathway. Treatment with 100 nM AngII increases the BRET signal with a peak before one minute, a sign of activation of the GαoA pathway (Figure 5A,B). Neither the toxin nor the candesartan alone can induce activation of this pathway (Figure 5A). The activation is blocked by 100 µM of toxin as well as 1 µM candesartan (Figure 5B). Thus, the toxin seems to display no agonist mode of action but an antagonist mode of action on the GαoA pathway. AngII induces GαoA activation in a dose-dependent manner with an EC_50_ of 12.5 ± 5.0 nM (*n* = 3) (Figure 5C). In the presence of the toxin at 150 µM the dose-response curve of AngII is shifted towards higher concentrations and its EC_50_ goes from 12.5 ± 5.0 nM to 966 nM ± 470 nM (*n* = 3) (Figure 5C). However, the maximal effect of AngII is not impaired by the toxin, indicating that A-CTX-cMila is also acting as a competitive antagonist on the GαoA pathway.

To conclude on the effect of A-CTX-cMila on the G-protein dependent pathways, this toxin behaves as a competitive antagonist on Gαq/PLC, Gαi3 and GαoA pathways.

### 2.6. Pharmacological Characterization of A-CTX-cMila on AT1-Mediated β-arrestin 2 Recruitment

Once G-proteins are activated, the AT1 receptor undergoes desensitization through its phosphorylation that triggers the binding to β-arrestins. For the recruitment of β-arrestin by the AT1 receptor in response to AngII stimulation, HEK293T cells were co-transfected with Rluc-8-βarrestin 2 and AT1-YFP expression plasmids [37]. An increase in the BRET signal is a sign of β-arrestin 2 recruitment upon AT1 activation.

The kinetic curve shows that 10 nM AngII induces an increase in the BRET signal, the signal reaching a peak at 3 min before stabilizing (Figure 6A,B). Neither the candesartan nor the toxin alone could induce an increase in the BRET signal (Figure 6A). In the presence of AngII, 100 µM of A-CTX-cMila completely inhibits the BRET signal as well as 1 µM candesartan (Figure 6B). Thus, the toxin displays no agonist mode of action but an antagonist mode of action on this pathway. AngII induces the β-arrestin 2 recruitment with a dose-dependent effect (EC_50_ = 0.99 ± 0.24 nM, *n* = 5) (Figure 6C). In the presence of 30 nM AngII, the toxin inhibits β-arrestin 2 recruitment by AngII in a dose-dependent manner with a micromolar potency (IC_50_ = 15.6 ± 3.0 µM, *n* = 4) (Figure 6D). The analysis of the dose-response curves of AngII submitted to different doses of toxin underlines that the EC_50_ of AngII raises from 0.99 ± 0.24 nM to 10.9 ± 5.8 nM with 15 µM toxin, to 23.3 ± 10.7 nM with 50 µM toxin, to 54.0 ± 19.0 nM with 150 µM toxin without affecting the maximal response of AngII by the toxin (*n* = 3) (Figure 6E). Schild’s regression is linear with a slope of −1.0 ± 0.4 and a pA2 of 5.7 ± 0.3 (*n* = 3). The associated K_B_ is 4.1 ± 3.1 µM (*n* = 3) (Figure 6F). A-CTX-cMila is also a competitive antagonist in the recruitment of β-arrestin 2.

### 2.7. Pharmacological Characterization of A-CTX-cMila on AT1-Mediated ERK_1/2_ Activation

AngII activates ERK_1/2_ mitogen-activated protein kinase (MAPK) pathway through both G protein and β-arrestin-dependent processes.

A-CTX-cMila, showing antagonist properties on these pathways, should inhibit downstream pathways and thus ERK_1/2_ activation by AngII. The activation level of ERK_1/2_ was determined with an HTRF assay based on the use of an antibody recognizing the phosphorylated/activated form of these kinases proteins. The kinetic study shows a rapid increase in ERK_1/2_ phosphorylation in presence of 30 nM of AngII, this phosphorylation peaking around 3 min and then stabilizing to an intermediate level until 15 min. In the presence of 150 µM A-CTX-cMila, ERK_1/2_ phosphorylation is strongly inhibited, demonstrating that A-CTX-cMila is an antagonist in ERK_1/2_ activation (Figure 7A).

To confirm the results obtained by the HTRF method, the effect of AngII and the toxin were determined by the western blot approach (Figure 7B,C). The results are in agreement with those obtained with the HTRF assay and show that AngII (10 nM) induces the phosphorylation of both ERK_1_ and ERK_2_ proteins and that A-CTX-cMila (20 µM) strongly reduces this effect (89% ± 2% inhibition).

### 2.8. Selectivity Profile of A-CTX-cMila

The selectivity of A-CTX-cMila was evaluated on other peptide-activated GPCRs involved in cardiovascular functions: endothelin type A (ETA), endothelin type B (ETB), apelin (APJ), vasopressin type 1a (V1a) and vasopressin type 2 (V2) receptors.

The activation of these receptors was tested on HEK293T cells transiently transfected with each receptor. For Gαq/PLC coupled receptors (ETA, ETB, V1a), IP1 production was quantified. For the V2 receptor, coupled to Gαs/adenylate cyclase pathway, cAMP production was analyzed. For the apelin receptor APJ, which is coupled with Gαi proteins, HEK293T cells were co-transfected with the APJ receptor together with Gαqi9 chimeric protein, which allows the coupling of Gαi-coupled receptors to PLC [38]. For this receptor, IP1 production was thus also quantified. On transfected HEK293T cells, IP1 or cAMP production is measured by HTRF assay kits in the presence of the cognate endogenous agonists and/or the toxin. 100 µM of A-CTX-cMila has no impact on IP1 production both in the presence and absence of ET-1, AVP or apelin. The toxin thus exhibits no agonist or antagonist effect on ETA, ETB, APJ and V1a receptors. Furthermore, 100 µM of A-CTX-cMila has no impact on cAMP production both in the presence and absence of AVP: the toxin exhibits no effect on the V2 receptor (Figure 8).

## 3. Discussion

Targeting the cardiovascular system is one of the strategies selected during the evolution of venomous animals to subdue their prey or defend themselves against predators. A significant number of toxins with cardiovascular effects have been identified and characterized. Among them, toxins targeting selective subtypes of sodium, potassium or calcium channels can directly affect cardiovascular functions. For example, toxins from scorpions, sea anemones or spiders may disrupt the Na_v_1.5 or hERG functions, leading to heartbeat rate variation or delay in cardiomyocyte repolarization, respectively [39]. Other snake toxins displaying phospholipase A2 activity can induce a decrease in blood pressure through the production of arachidonic acid [40]. Among the GPCRs-interacting toxins, Atractaspis snakes sarafotoxins activate endothelin receptors, inducing a strong general vasoconstriction leading to heart failure and cardiac arrest [41]. Finally, the M2 muscarinic toxin MT9 from the black mamba increases mesenteric artery contraction [42]. Despite the existence of many toxins with cardiovascular effects, there is only one family of toxins known to target the RAS: these are the bradykinin-potentiating peptides isolated from Brazilian pit viper which led to the development of the antihypertensive blockbuster captopril targeting ACE [43]. Nevertheless, there are no identified toxin targeting GPCRs involved in the RAS.

The aim of the present study was to find GPCR-interacting toxins targeting the RAS. To this end, we have screened the Venomics library and identified A-CTX-cMila from the venom of *Conus miliaris* as the first toxin active on the AT1 and AT2 receptors. It interacts at high nanomolar range affinity on the AT1 receptor and displays an antagonist mode of action on all the signaling pathways studied. Indeed, the toxin was found to be a competitive antagonist on G-protein dependent and independent pathways coupled to the AT1 receptor: it inhibits Gαq, Gαi3, GαoA activation, β-arrestin 2 recruitment and blocks ERK_1/2_ activation. A-CTX-cMila was also found to bind to the AT2 receptor but with an about 100-fold lower affinity than for the AT1 receptor. It also inhibits the binding of fluorescent forms of AngII (Cy3 and Cy5 conjugates) to intact cells. Considering the low affinity of A-CTX-cMila for the AT2 receptor, and the difficulties of characterizing AT2-dependent signaling pathways [44] the functional study of the toxin was only performed on the AT1 receptor.

The a-CTX-cMila sequence (CHFWVCP) is very different from that of AngII (DRVYIHPF). The most important residues for AngII activity on the AT1 receptor are Tyr4, His6 and Phe8. Tyr4 interacts with Asn111, also determining the switch for full-agonist activity [45]. His6 interacts with Phe259, Thr260 and Asp263, thus providing the docking site [46]. Phe8 interacts with Lys199 and His256, critical for Gαq signaling [47]. Thus, considering the absence of sequence homology with AngII, and the cyclic structure of the toxin, it is unlikely that they share a similar interaction mode, in accordance with its antagonism mode of action. This is confirmed by the implication of other residues of AT1 in the binding with an AT1 receptor antagonist, olmesartan, which are Tyr35, Trp84, and Arg167 [30]. In order to assess the specificity of this interaction, the effect of A-CTX-cMila was also tested on other receptors involved in cardiovascular homeostasis and having endogenous peptide ligands. The apelin receptor (APJ) is one of the closest receptors to AT1 in the phylogenetic tree of GPCRs [48] with about 30% homology in humans [49]. It is an important regulator of body fluid homeostasis and cardiovascular functions [50]. This receptor has two unrelated families of endogenous peptide ligands, apelin and elabela [37], which are both linear peptides devoid of Cys residues [51]. ETA and ETB endothelin receptors have been detected predominantly in cardiovascular tissues. Endothelins are 21 amino acid peptides with four Cys residues forming two disulfide bridges [52]. ETA receptor is located mostly in vascular smooth muscle cells (VSCM) and promotes vasoconstriction but also inflammation and cell proliferation [53]. ETB receptor is mainly expressed on endothelial cells, where it promotes vasodilatation through the production of nitric oxide. Finally, the V1a receptor is a Gαq/PLC coupled receptor expressed in vascular smooth muscle cells (VSMC) increasing vasoconstriction when activated by arginine-vasopressin (AVP) [54]. The V2 receptor is found on the basolateral surface of the cells of the collecting ducts. It is a Gαs-coupled GPCR involved in diuresis regulation whose activation by its endogenous agonist ligand, AVP, leads to the homeostatic regulation of water and sodium recapture from the urine to the bloodstream and an anti-diuretic effect [55]. The results obtained show that A-CTX-cMila, has neither agonistic nor antagonistic effect on these receptors, indicating that the toxin action is restricted to angiotensin II receptors.

Conotoxins represent highly specific biological probes that provide tools for pharmacological studies. So far, less than 1% of the conotoxin peptide library has been cataloged [34]. The cyclic nature of A-CTX-cMila is not common in the conotoxins family and is found only in contryphans and in conopressins families. Contryphans target calcium channels [56], have no sequence homology with A-CTX-cMila and possess a loop of 5 residues [57]. Conopressins are related to vasopressin and oxytocin, endogenous peptides that bind vasopressin and oxytocin receptors respectively, and act on these same receptors. Like the A-CTX-cMila, conopressins have a four-residue loop such as conopressin-T (CYIQNCLRV) from *Conus tulipa,* conopressin-S (CIIRNCPRG) from *Conus striatus* [58] and conopressins M1 (CFPGNCPDS) and M2 (CFLGNCPDS) from *Conus miliaris* [59]. However, they display no sequence homology with A-CTX-cMila and no similarity in their pharmacological profiles. Indeed, A-CTX-cMila had no effect on vasopressin receptors. These results demonstrate that A-CTX-cMila is the first member of a novel group of cyclic conotoxin associated with a new function.

It is interesting to note that our study was performed on mammalian receptors, while *Conus miliaris* is a worm hunter. If this experimental bias may account for the moderate affinities of the A-CTX-cMila on AT1 and AT2, it may reveal a biological role of the toxin for predation and/or protection. Indeed, GPCR functions in worms are highly developed. Over the past decade, more and more RAS effectors have been discovered in invertebrates, such as angiotensinogen-like peptides, and renin-like and ACE-like enzymes. In clam worms, an ACE-like form was discovered, displaying similar properties as mammalian ACE [60]. Physiological roles of the mammalian RAS may also be encountered in invertebrates and more specifically in worms. For instance, it has been demonstrated that AngII and AngIII had an impact on body fluid volume in the clam worm [61]. We may consider that the toxin we isolated from *Conus miliaris* may act through the worm’s RAS system to apprehend it. This work identified a possible new strategy of predation and/or protection used by marine cone snails that deserved to be deeply studied.

A-CTX-cMila is part of the Venomics databank and contains 20,206 sequences including 606 cyclic toxins [62]. The biological function of this class of toxins is poorly studied, despite a fairly easy production, and therefore represents high potential study tools. A-CTX-cMila can be seen as the spearhead of a new axis of study of cyclic conotoxin with high pharmacological potential.

## 4. Materials and Methods

### 4.1. Cell Culture and Transfection

For the establishment of CHO-AT1a-EGFP and CHO-AT2-EGFP stable cell lines, the sequence of the rat AT1a (gift from Dr. K. Bernstein [63]) and AT2 [64] receptor cDNAs were amplified by PCR using oligonucleotide primers allowing the deletion of the stop codon and the in-frame cloning in pEFGP-N1 plasmid (Clontech) in HindIII and BamHI restriction sites. The constructs were checked by sequencing. CHO-K1 cells (from ATCC) were transfected with 1 µg of the plasmids and selected in the presence of 750 µg/mL G418 (GIBCO, ThermoFischer Scientific). For final selection, GFP-positive cells were sorted by FACS analysis on an Epics EST flow cytometer equipped with an Autoclone cell sorter (Beckman Coulter).

CHO-AT1a-EGFP, CHO-AT2-EGFP and HEK293T (from ATCC) cells were maintained at 37 °C and 5% CO_2_ in DMEM/F-12 medium supplemented with 10% fetal bovine serum (FBS), penicillin (50 U/mL), streptomycin (50 μg/mL) and glutaMAX-CTS (0.5 mM). HEK293T cells were transfected with Lipofectamine™ 2000 reagent according to the manufacturer’s specifications. All reactants were supplied from Gibco, ThermoFischer Scientific. KB-1753-Nanoluc, Gαi3-YFP and GαoA-YFP were obtained from M. Maziarz (Department of Biochemistry, Boston University School of Medicine, Boston, MA 02118, USA).

### 4.2. Membrane Preparation

Transfected cells (15 × 10^6^ cells) were washed with 10 mL of phosphate-buffered saline (PBS) and 0.5 mM EDTA, pH 7.4 and harvested using 10 mL of this solution. Cells were collected by centrifugation at 2000× *g* for 3 min at 4 °C. Cells were homogenized in 5 mL of 10 mM HEPES buffer (pH 7.4) at 4 °C containing 70 μL anti-protease cocktail (AEBSF at 104 mM, aprotinin at 80 μM, bestatin at 4 mM, E-64 at 1.4 mM, leupeptin at 2 mM and pepstatin A at 1.5 mM, Sigma Aldrich, Burlington, MA, USA), using a Potter’s grinder and then centrifuged at 1000× *g* for 10 min at 4 °C. The resulting supernatant was then centrifuged at 50,000× *g* for 1 h at 4 °C. The crude membrane pellet was washed and resuspended in 1 mL of 10 mM HEPES (pH 7.4).

### 4.3. Radioligand Binding Assays

Sar1, Ile8-AngII (30 µg in TRIS pH 7.5; 100 mM) was introduced in a tube coated with iodogen (ThermoFischer Scientific) with 0.4 mCi of Na^125^I (Perkin Elmer, Waltham, MA, USA, specific activity 17 Ci/mg, 0.1 M NaOH) for 20 min. After purification by HPLC and mass control with MALDI-TOF, the monoiodinated peptide was diluted in 10 mM HEPES supplemented with 1% bovine serum albumine (BSA).

Binding experiments were performed at room temperature using a mix of 100 μL of radioligand at 1 nM in 50 mM Tris HCl pH 7.4, 1 mM MgCl_2_, 0.1 g/L BSA, 0.3 μL of membranes of cells expressing receptors, and the competitor at different concentrations. Non-specific binding was determined in presence of 2 μM AngII. Filter plate (Unifilter−96 GF/C, Perkin Elmer) was saturated with polyethylenimine (PEI, 0.5%, Sigma Aldrich, Burlington, MA, USA). After 16 h of incubation at room temperature, samples were filtered and 25 μL of microscintillant (MicroScint-O, Perkin Elmer) were added to each dry filter before recording the radioactivity on the Packard TopCount counter (Perkin Elmer).

The toxins library was produced by the Venomics consortium [1]. The screening conditions (quantity of membranes, quantity of radioligand) were optimized beforehand to obtain a Z factor greater than 0.5, with Z = 1 − ((3 × MAD_max_ + 3 × MAD_min_)/(median_max_ − median_min_)), where MAD = Median absolute deviation. Eleven 96-well plates containing one toxin by well were screened. On each 96-well plate, eight wells were dedicated to total binding of the radioligand, and eight others were dedicated to non-specific binding. After counting the radioactivity bound in each well, peptides with an associated binding value less than Median – 3 × MAD were classified as hits.

### 4.4. Production of A-CTX-cMila

The protected amino acids (AA) came from Novabiochem and Activotec; Fmoc-Pro-Novasyn TGT resin, dichloromethane, acetic anhydride (Ac_2_O) from Merck Millipore; N-methylmorpholine (NMM), thioanisole, anisole, triisopropylsilane (TIPS), HCTU, piperidine, acetonitrile (CH_3_CN), HEPES, cysteine, cystine from Sigma Aldrich; acetic acid, N-Methyl-2-pyrrolidone (NMP), methanol from VWR; diethyl ether, dimethylformamide (DMF) from Carlo Erba; trifluoroacetic acid (TFA) from Fisher Scientific.

SPPS was performed on a synthesizer (Prelude Protein Technologies Inc, Tucson, AZ, USA, Prelude User software) using Fmoc-strategy. Resin was swollen with 2 × 3 mL of DCM for 30 s and then with 1 × 3 mL of NMP for 5 min. The Fmoc group was deprotected with 2 × 2 mL of 20% piperidine in NMP for 2 min, followed by 3 × 3 mL of NMP for 30 s. Coupling steps were completed twice for 10 min with 1.3 mL of AA (200 mM in NMP, cysteine in DMF) added to 1 mL of HCTU (250 mM in NMP) and 0.5 mL of NMM (1 M in NMP), followed by 2 × 3 mL of NMP for 30 s. Capping was carried out with 2 mL 250 mM Ac_2_O in NMP for 5 min. A final washing of the medium with NMP (3 × 3 mL for 30 s) was carried out.

The peptide was cleaved from its solid support and the side chains were deprotected with a solution of TFA 82.5%/Thioanisole 5%/Anisole 5%/H_2_O 5%/TIPS 2.5% (10 mL; 2 h at room temperature; 300 rpm stirring). The peptide in solution was then precipitated in 3 × 40 mL of cold diethyl ether (3000 rpm for 4 min), dissolved in 15 mL of 10% acetic acid and freeze-dried.

### 4.5. HPLC Purification and Analysis

The mobile phases that were used for the HPLC analysis are A: H_2_O + 0.1% TFA and B: CH_3_CN + 0.1% TFA. The crude peptide is purified by a preparative method on a Waters device (pump module: 2535 Quaternary Gradient module; UV detector: Waters 2998 Photodiode Array), with an XBridge BEH 300A Prep C18 column OBD 5 μm 19 × 250 mm, at a flow rate of 13 mL/min. Analytical HPLC is performed on a Waters device (pump module: e2695 separation module; UV detector: Waters 2998 Photodiode Array) with an XBridge BEH 300A C18 column (5 μm, 4.6 mm × 250 mm, Waters). The indicated retention times are given in minutes. Gradient used was 0–50% B in 50 min with a wash phase at 80% B for 10 min and a column equilibration phase at 100% A for 10 min.

### 4.6. Mass Spectrometry Analysis

Samples were analyzed by HPLC coupled with mass spectrometry (HPLC-MS): the electrospray ionization mass spectrometry technique of the ESI-IT ion trap type (ElectroSpray Ionization-Ion Trap) on an Esquire HCT device (high capacity trap, Bruker Daltonics) was coupled with reversed-phase chromatographic analysis (Agilent 1100 series HPLC), with an Agilent Eclipse XDB C18 80A column, 5 μm, 4.6 mm × 150 mm. The gradient used was 5 to 100% B in 60 min at 1 mL/min, with a wash phase at 80% B for 10 min and a column equilibration phase at 95% A for 10 min. The data were analyzed with DataAnalysis software (Bruker Daltonics). The indicated retention times were given in minutes. Samples were also analyzed by mass spectrometry MALDI-TOF (Matrix-assisted laser desorption ionization Time of flight), using a 4800 MALDI TOF-TOF Analyzer (Applied Biosystems), in reflectron mode. The sample was co-crystallized on a plate with a solution of α-cyano-4-hydroxy-cinnamic acid (HCCA), solution prepared at 10 mg/mL in 50/50 CH_3_CN/H_2_0, 0.1% TFA. The reported m/z values correspond to the monoisotopic peak.

### 4.7. Thermodynamic Oxidation

Oxidation buffer composed of HEPES (pH 7.5; 0.1 M in H_2_O), EDTA (pH 8; 0.001 M in H_2_O), cysteine (0.001 M in H_2_O), cystine (0.0001 M in H_2_O), CH_3_CN (20%) and purified linear peptide (20 μM in H_2_O), was mixed at room temperature for 48 h.

### 4.8. Labeling of AngII with Cy3 and Cy5

The Cy3 fluorophore (4 eq; 4 μmol in NMP, ThermoFischer Scientific) was mixed with HCTU (4 eq; 4 μmol) and NMM (10 eq; 10 μmol) in NMP. Notably, 1 μmol of peptide on resin was added (300 rpm stirring; 2 h; room temperature). Notably, 3 × 5 mL washes with NMP then 3 × 5 mL washes with DCM were carried out. The peptide on resin was dried under vacuum, cleaved from the resin and deprotected.

0.3 μmol of peptide cleaved from the resin, deprotected, purified and freeze-dried was taken up in DMF. N,N-diisopropylethylamine (DIPEA, Sigma Aldrich, 10 eq.) was added, then the Cy5 fluorophore (Lumiprobe, 5 eq.), with stirring at 250 rpm for 1.5 h at 37 °C. TRIS (pH 7.8; 50 mM) was added (250 rpm stirring; 20 min).

Both Cy3-AngII and Cy5-AngII were purified by HPLC and analyzed by mass spectrometry. After lyophilization, peptides were resolubilized in H_2_O.

### 4.9. Fluorescent-AngII Binding Assays

CHO-AT1a-EGFP and CHO-AT2-EGFP were seeded at 40,000 cells/well in 96-well microplate. After 24 h, cells were preincubated or not with 50 µM toxin for 30 min and then incubated with 10 nM Cy3-AngII for 15 min at 37 °C. Cells were then washed with Ham’s F-12 and fixed with 4% PFA (Sigma Aldrich) for 5 min. Vectashield-DAPI was added (Sigma Aldrich) to each well before observation. Microscopic observation was made under a Nikon Eclipse Ti microscope equipped with a 60× magnification objective.

For flow cytometry analysis, CHO-AT1a-EGFP or CHO-AT2-EGFP cells (200,000 cells) were incubated in suspension in PBS containing 0.1% BSA in the presence or absence of 150 µM toxin for 30 min before addition of 20 nM Cy5-AngII for 30 min at room temperature in a total volume of 100 µL. Samples were diluted ten-fold in PBS-BSA immediately before analysis of an ACCURI C6 + flow cytometer (BD Bioscience) equipped with 488 and 540 lasers.

### 4.10. Second Messengers Assays

Experiments were performed on CHO cells stably expressing receptors of interest, or on HEK293T cells transiently transfected. In this case, HEK293T cells were seeded in 6-well plates (10^6^/well) and transfected 24 h later with Lipofectamine™2000. Notably, 24 h after transfection, cells were collected and resuspended in appropriated buffer (see below).

#### 4.10.1. IP1 Assay

IP1 production was quantified with the IP-one-Gq assay kit (Cisbio Perkin Elmer) based on homogeneous time-resolved fluorescence (HTRF) according to the kit protocole. Cells expressing the receptor of interest were added to the 384-well microplate (22,500 cells in 5 µL stimulation buffer) and stimulated for 45 min at 37 °C in the presence of agonist or toxin or by a mix of agonist and toxin prepared in stimulation buffer. In this latter case, the toxin was pre-incubated with cells for 30 min at 37 °C. Detection reagents (2 × 5 µL) were then added to each well and the plate was incubated for 30 min at room temperature before reading on a CLARIOstar plate reader (BMG Labtech, Ortenberg, Germany).

#### 4.10.2. cAMP Assay

Cyclic AMP production was quantified with the cAMP Gs dynamic kit (Cisbio Perkin Elmer) based on HTRF. Cells expressing V2 receptor were added to the 384-well microplate (1500 cells/well in 5 µL Ham’s F-12 medium containing 1 mM IBMX) and stimulated for 45 min at 37 °C by 30 nM vasopressin or 100 μM toxin or by a mix of vasopressin and toxin. Detection reagents (2 × 5 µL) were then added to each well and the plate was incubated for 30 min at room temperature before reading on a CLARIOstar plate reader (BMG Labtech).

### 4.11. β-arrestin 2 Recruitment and Biosensors Assays

For the β-arrestin 2 recruitment assay, HEK293T cells seeded in 6-well plates (10^6^/well) were transfected with Lipofectamine™ 2000 with 2.5 μg of DNA mix containing 80% of Rluc8-β-arrestin 2 plasmid (BRET donor) and 20% AT1-YFP plasmid (BRET acceptor). Control cells with no BRET acceptor were made by transfecting cells with Rluc8-β-arrestin 2 plasmid and non-fluorescent AT1 plasmid. Notably, 24 h after transfection, cells were collected and resuspended in reaction buffer (HEPES 25 mM pH 7.5; CaCl_2_ 1 mM; MgCl_2_ 0.5 mM; NaCl 150 mM; glucose 1 g/L) and 20 μL of cell suspension (40,000 cells) were placed in a 96-well plate (Perkin Elmer; OptiPlate-96, White Opaque 96-well Microplate). Coelantherazine H (Nanolight technology, Pinetop) (20 μL of a 5 μM solution prepared in reaction buffer) was added to each well before the addition of agonist ligands (10 μL in reaction buffer). Antagonist molecules and toxins were pre-incubated with the cells for 30 min at 37 °C before adding the substrate and the agonist ligand. Luminescence signals were measured with an Envison microplate reader (Perkin Elmer, 2104 multi-label reader) set at 37 °C with 485-nm and 530-nm emission filters. The BRET signal was calculated as the 530 nm/485 nm signal ratio. The net BRET signal was determined by subtracting the BRET signal obtained in the absence of acceptor. Ligand-induced BRET is calculated by subtracting the net BRET before stimulation from the net BRET after stimulation. For kinetic BRET measurements, signals were measured every 1 s for the duration of the experiment.

For Gαi3 and GαoA activation assays, HEK293T cells were transfected as described above but with DNA mix containing 0.016% of KB-1753-Nanoluc plasmid (BRET donor) 40% of Gαi3-YFP or GαoA-YFP plasmid (BRET acceptor) and 60% non-fluorescent AT1 plasmid. Notably, 24 h after transfection, cells were collected and resuspended in OptiMEM medium (ThermoFischer Scientific). Stimulation of cells and luminescence recordings were made for β-arrestine2 recruitment except that cells were resuspended at 25,000 cells in 20 µL and that the substrate solution consisted of 20 µL of 6.25 μM furimazine (Promega corporation).

### 4.12. ERK_1/2_ Activation

ERK_1/2_ phosphorylation was assessed with the Cisbio Phospho-ERK_1/2_ (Thr202/Tyr204) cellular kit. CHO-AT1a-EGFP cells were seeded 48 h before at 40,000 cells/well in a 96-well plate. The medium was changed 24 h after seeding and replaced with Ham’s F-12 medium without FBS. At the time of experiment, culture medium was discarded and replaced by 40 µL Ham’s F-12 medium containing toxin or not, and the plate was incubated for 30 min at 37 °C. Cells were stimulated by 10 µL of 5× AngII solution (10 μL/well; 30 nM) in Ham’s F-12 medium, at the desired times (0, 1, 2, 3, 4, 5, 6, 7, 10, 15 min). The wells were emptied and then 50 μL of lysis buffer was added. The plate was incubated for 1 h at room temperature. Notably, 16 μL of each well was transferred to a 384-well plate. The reagents (4 µL, 1:1) were added, and the plate was incubated for 3 h at room temperature before reading with the CLARIOstar. For western blot analysis of phosphorylated ERK_1/2_, CHO-AT1a-EGFP cells were seeded in 12-well plates at a density of 200,000 cells/well. Medium was replaced by F12 medium without FCS 24 h before the experiment. Cells were incubated or not with 20 µM A-CTX-cMila for 30 min, 10 nM AngII were then added for 5 min and cells were lysed in 100 µL ice-cold lysis buffer (PBS, 1% Triton X-100). Detergent-extracted proteins were analyzed by western blot technique using mouse monoclonal anti-active phosphorylated ERK_1/2_ (Cell Signaling Technology) and rabbit polyclonal anti-ERK2 (Santa Cruz Biotechnology) antibodies (1:5000 each). The membranes were then incubated with anti-rabbit IgG antibody conjugated to IRDye 800CW (LI-COR Biosciences) and anti-mouse IgG antibody conjugated to AlexaFluor 680 (Life technologies) for 1 h at 37 °C. Protein bands were detected and quantified with an Odyssey Infrared Imaging System (LI-COR Biosciences).

### 4.13. Data Analysis

All experiments were performed at least three times, as indicated in the figure legends, and, unless otherwise indicated, they are presented as averages ± SEM. Binding experiments, second messengers assays, and BRET assays were all analyzed using Prism v5.000 software (GraphPad). Dose−response curves were generated using nonlinear regression curves with three parameters. Schild’s regressions were generated by plotting log(A), with A = (EC50 _AngII + toxin_/EC50 _AngII_) − 1 depending on the −log(concentration ligand) using simple linear regression curves. The *K*_B_ value was calculated from the *y*-intercept of the slope (*K*_B_ = 10^−*y*-intercept^). For multiple comparison tests, statistical analysis was performed using one- or two-way ANOVA followed by post hoc comparisons with Tukey’s HSD test. Values with *p* < 0.05 were considered statistically significant.

## Figures and Tables

**Figure 1 ijms-24-02330-f001:**
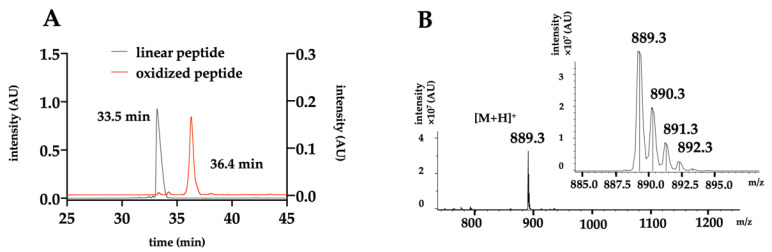
HPLC and mass analysis of the toxin. (**A**) Analytical HPLC of the linear and oxidized forms of A-CTX-cMila, with a linear gradient of acetonitrile at 1%/min. The results show superimposed UV chromatograms at 280 nm of the purified linear peptide in black (left *y* axis) and the purified oxidized peptide in red (right *y* axis). (**B**) ESI-MS spectrum of the purified A-CTX-cMila in positive ESI mode of ionization. The inset shows the isotopic distribution. All chromatograms and spectra are representative of all the batches produced.

**Figure 2 ijms-24-02330-f002:**
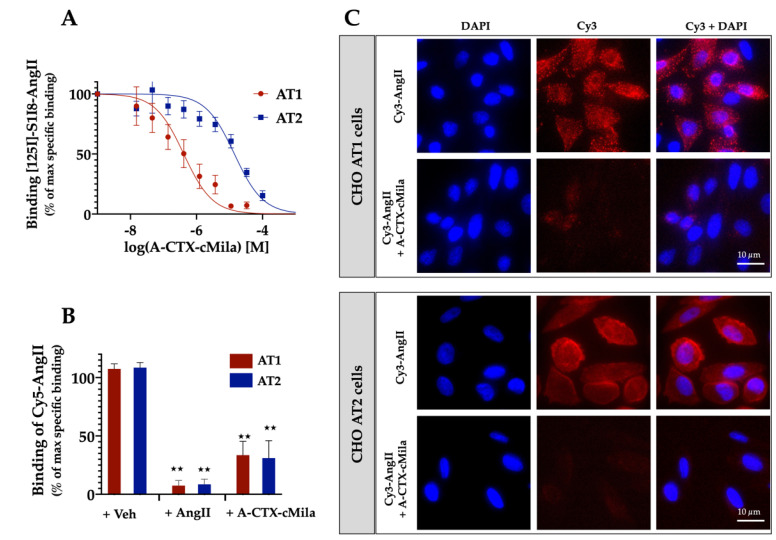
Binding properties of A-CTX-cMila on angiotensin II receptors. (**A**) Competition binding curves of ^125^I-Sar1-Ile8-AngII on membranes of HEK293T cells expressing AT1 or AT2 receptors in presence of increasing concentrations of A-CTX-cMila. Data are expressed as % of ^125^I-Sar1-Ile8-AngII specific binding in the absence of toxin. Values are means (%) ± SEM (*n* = 5). (**B**) Binding of Cy5-AngII (20 nM) on CHO cells expressing AT1 or AT2 receptors in the absence or presence of A-CTX-cMila (100 µM) or AngII (1 µM). Cells were incubated with the toxin for 30 min before addition of Cy5-AngII. Binding of Cy5-AngII was analyzed by flow cytometry. Values are expressed as percentages of the specific labeling and given as mean (%) ± SEM (*n* = 4). (**C**) Labelling of CHO cells expressing AT1 or AT2 receptors by Cy3-AngII (10 nM) in the absence or presence of A-CTX-cMila (50 µM). Cells were incubated with the toxin for 30 min before addition of Cy3-AngII. Images are representative of three independent experiments. Veh: vehicle; ** *p* < 0.0001, significantly different compared to vehicle.

**Figure 3 ijms-24-02330-f003:**
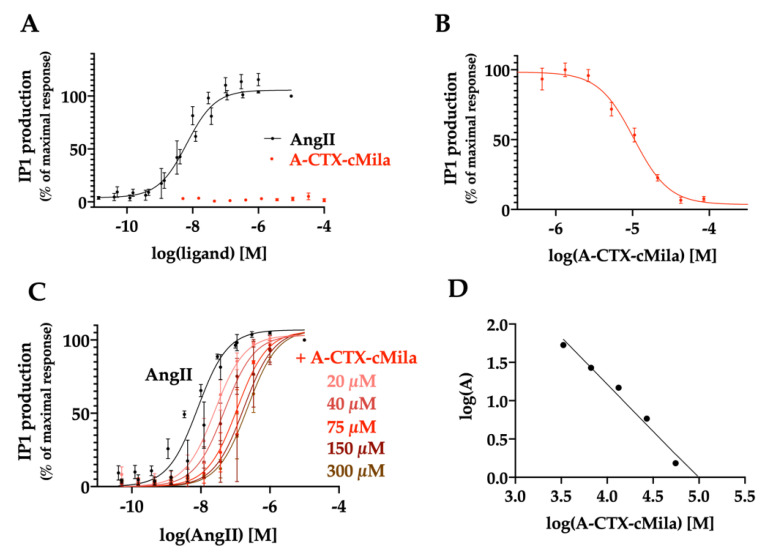
Competitive antagonist mode of action of A-CTX-cMila on AT1-mediated Gαq/PLC pathway on CHO-AT1a-EGFP cells. (**A**) Effect of AngII (*n* = 6) and A-CTX-cMila on IP1 production (*n* = 3). Cells were incubated with increasing concentrations of AngII or the toxin for 45 min. (**B**) Effect of the toxin on IP1 production in presence of 10 nM AngII (*n* = 3). (**C**) Impact of different doses of toxin on the dose-dependent effect of AngII (*n* = 3). For (**B**,**C**), cells were incubated with increasing concentrations of toxin for 30 min before addition of AngII for 45 min. In (**A**–**C**), IP1 production was assessed using HTRF assay kit. Values are expressed as a percentage of the maximal AngII response and are mean ± SEM. (**D**) Schild’s regression of EC_50_ values was deduced from the experiment shown in (**C**). Representation of one typical experiment among three performed.

**Figure 4 ijms-24-02330-f004:**
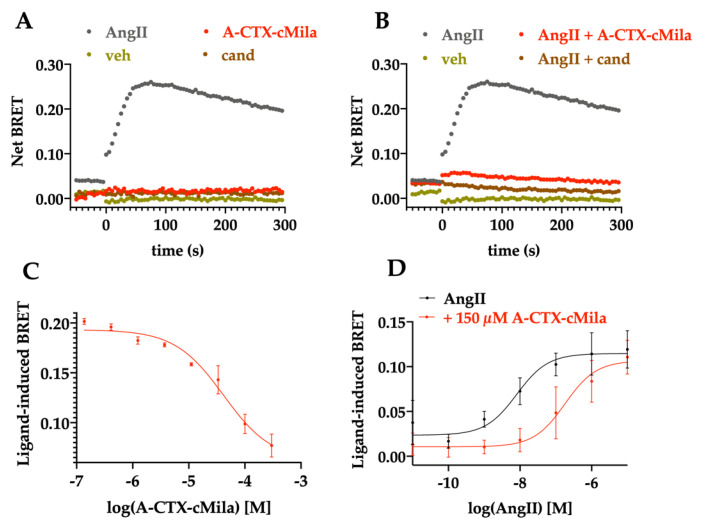
Competitive antagonist mode of action of A-CTX-cMila on AT1-mediated Gαi3 pathway monitored by BRET biosensors assays. HEK293T cells were transfected with plasmids encoding AT1 receptor, KB-1753-Nanoluc and Gαi3-YFP proteins and processed for BRET recordings. (**A**,**B**) Kinetic curves of Gαi3 activation in presence of 10 nM AngII in the absence or presence of 100 µM A-CTX-cMila or 1 µM candesartan. Representation of one typical experiment among three independent experiments. (**C**) Impact of A-CTX-cMila on Gαi3 activation in presence of 50 nM AngII. Representation of one typical experiment among three independent experiments. (**D**) Impact of 150 µM of A-CTX-cMila on the dose-dependent effect of AngII on Gαi3 activation. Values are mean ± SEM (*n* = 3). For (B), (C) and (D), cells were incubated with increasing concentrations of toxin or candesartan for 30 min before addition of AngII at t = 0. Veh: vehicle; cand: candesartan. Net BRET and ligand-induced BRET were calculated as described in “Material and Methods” section.

**Figure 5 ijms-24-02330-f005:**
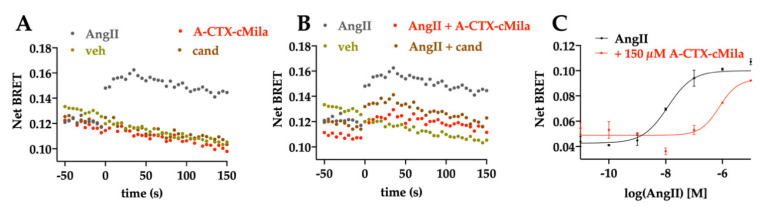
Competitive antagonist mode of action of A-CTX-cMila on AT1-mediated GαoA pathway monitored by BRET biosensors assays. HEK293T cells were transfected with plasmids encoding AT1 receptor, KB-1753-Nanoluc and GαoA-YFP and processed for BRET recordings. (**A,B**) Kinetic curves of GαoA activation in presence of 100 nM AngII in the absence or presence of 100 µM A-CTX-cMila or 1 µM candesartan. Representation of one typical experiment among three independent experiments. (**C**) Impact of 150 µM of A-CTX-cMila on the dose-dependent effect of AngII on GαoA activation. Representation of one typical experiment among three independent experiments. For EC_50_, mean ± SEM (*n* = 3). For (**B**,**C**), cells were incubated with increasing concentrations of toxin or candesartan for 30 min before addition of AngII. Veh: vehicle; cand: candesartan. Net BRET and ligand-induced BRET were calculated as described in “Material and Methods” section.

**Figure 6 ijms-24-02330-f006:**
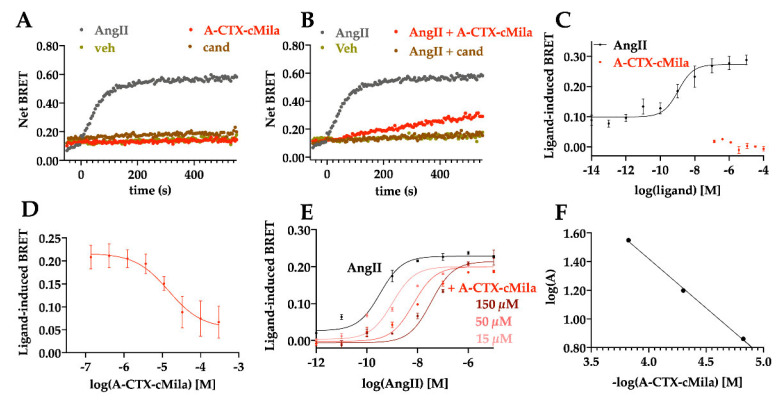
Competitive antagonist mode of action of A-CTX-cMila on AT1-mediated β-arrestin 2 recruitment. HEK293T cells were transfected with plasmids encoding AT1-YFP receptor and Rluc8-β-arrestin 2 and processed for BRET measurements. (**A**,**B**) Kinetic curves of β-arrestin 2 recruitment in presence of 10 nM AngII in the absence or presence of 100 µM A-CTX-cMila or 1 µM candesartan. Representation of one typical experiment among three independent experiments. (**C**) Effect of AngII and A-CTX-cMila on β-arrestin 2 recruitment. For AngII, mean ± SEM (*n* = 5), for A-CTX-cMila, representation of one experiment. (**D**) Impact of the toxin on β-arrestin 2 recruitment in presence of 30 nM AngII. Values are mean ± SEM (*n* = 4). (**E**) Impact of different doses of A-CTX-cMila on the dose-dependent effect of AngII on β-arrestin 2 recruitment. Representation of one typical experiment among three independent experiments, for EC_50_, mean ± SD (*n* = 3). For (**B**,**D**,**E**), cells were incubated with increasing concentrations of toxin or candesartan for 30 min before addition of AngII. (**F**) Schild’s regression of corresponding activation curves. Representation of one typical experiment among three independent experiments. Veh: vehicle; cand: candesartan. Net BRET and ligand-induced BRET were calculated as described in the “Material and Methods” section.

**Figure 7 ijms-24-02330-f007:**
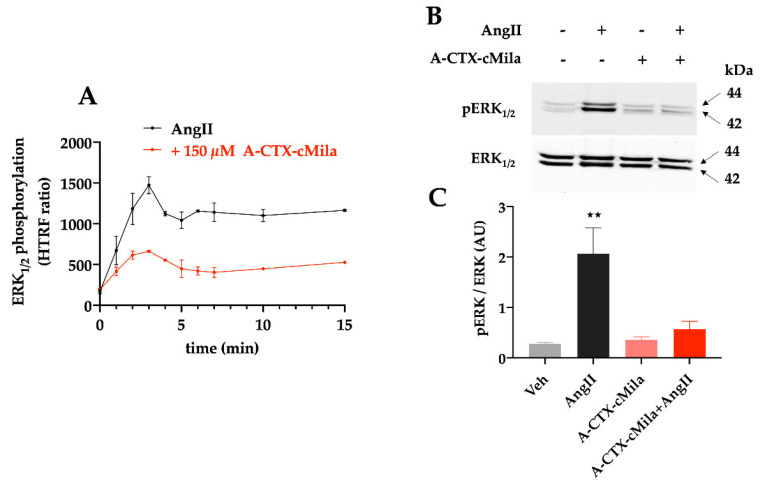
Antagonist mode of action of A-CTX-cMila on AT1-mediated ERK_1/2_ activation in CHO-AT1a-EGFP cells. (**A**) Time course of AT1-induced ERK_1/2_ phosphorylation by 30 nM AngII in the absence or presence of 150 µM A-CTX-cMila. ERK_1/2_ phosphorylation level was measured with HTRF kit. Representation of one typical experiment among three independent experiments, values are mean ± SD. (**B**) Western blot analysis of ERK_1/2_ phosphorylation. Cells were pre-incubated in the presence or absence of 150 µM A-CTX-cMila and then stimulated with 30 nM AngII for 5 min. Representation of typical results among three independent experiments. Notably, pERK_1/2_: phosphorylated ERK. (**C**) Quantification of western blots corresponding to (**B**). Values are expressed as the ratio of band intensities of phosphorylated ERK over total ERK and are mean ± SD (*n* = 4). ** *p* < 0.0001, significantly different compared to all other conditions tested.

**Figure 8 ijms-24-02330-f008:**
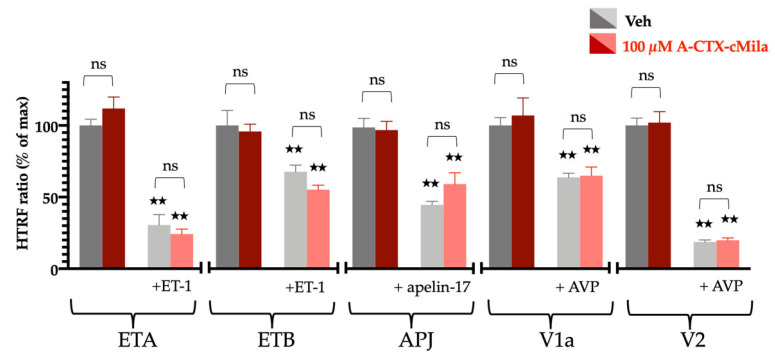
Selectivity profile of A-CTX-cMila for other GPCRs on HEK293T cells transfected with various receptors. Effect of 100 µM A-CTX-cMila on ETA, ETB, APJ and V1a-dependent production of IP1 or V2-dependent production of cAMP, in the presence or absence of ET-1 30 nM or apelin-17 100 nM or AVP 30 nM. For APJ receptors, cells were co-transfected with Gαqi9 expression plasmid. Values are expressed in HTRF ratio, mean ± SEM (*n* = 3). A decrease in the HTRF ratio is the sign of second messenger (IP1 or cAMP) concentration increase. Veh: vehicle; ns: non significantly different; ** *p* < 0.0001, significantly different compared to the corresponding Veh.

## Data Availability

Data sharing not applicable.

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
