# Peer review of "Characterization of the First Animal Toxin Acting as an Antagonist on AT1 Receptor"

_ijms, 2023, doi:10.3390/ijms24032330_

Round 1
Reviewer 1 Report
In this research article proposed for the Special Issue “Molecular Mechanisms of Animal Toxins, Venoms and Antivenoms”, the authors characterize a toxin (A-CTX-cMila) as a peptide capable of binding to AT1 receptor and perform as an antagonist, blocking several of the signaling pathways mediated by AT1R which apparently could be of some therapeutic interest.
Although the idea is interesting, as the paper is presented now it does not quality for acceptance. In my opinion some important controls are missing:
- While binding in Fig 2A shows the higher affinity of A-CTX-cMila for AT1R than for AT2R, the concentration used in most experiments is quite high (100-150 µM). What is the reason for choosing this dose? It is very likely that at this concentration the toxin can bind to AT2R and even to other receptors.
- Experiments of Fig 2B and 2C must be performed in cells expressing AT2R.
- Authors claim that “Even if AT2 receptor is classified as a GPCR and possesses all the GPCR common features, it seems to be unable to couple and activate any classical G-protein dependent pathways.”
This statement is wrong: several publications reports that AT2R bind to Gi protein (PMID: 8663053, PMID: 32807174; https://doi.org/10.1161/01.RES.87.9.753; DOI: 10.2741/3289).
- Pharmacological characterization of A-CTX-cMila on AT2R Gi mediated pathway should be performed. I recommend using some selective ligand for AT2R like C21 (DOI: 10.2174/1389203718666170203150344) or CGP-42112 (DOI: 10.1042/CS20220261).
- Fig 7: A bar graph with the quantification of the different western blot bands should be added.
- Update the discussion with the new data obtained for AT2R expressing cells.
- The possibility of binding of this conotoxin to the AT2R and/or other receptors not tested in Fig 8 experiments means that the potential therapeutic potential of blocking AT1R signaling is drastically reduced/limited.
Other corrections:
- Figure 2 legend: “membranes expressing AT1 and AT2”, membranes should express one receptor or the other, not both. Membranes obtained from?
- Figure 2C. Scale bar must be added.
- Fig 4C; Fig 5C; Fig 6C,E; Fig 7A. Some of the dots in the graphs lacks SD or SEM whiskers.
- In all figures, the experimental model used should be clear. Figure legends should be improved.
Reviewer 2 Report
Overall, the scientific content of this manuscript is strong and authors have done excellent work to characterize the bioactivity of this conotoxin. Below are minor suggestions and should be considered optional for publication.
1. The manuscript mentions that the peptide was isolated from the venom of C.miliaris. If the native material is available, performing a co-elution study (co-eluting the native peptide with the synthetic oxidized material) will strengthen the work. Could the authors comment on this?
2. Since the toxin is a competitive antagonist, has the research team tried pre-incubating the cells with the the toxin(for at least 5 minutes), before applying toxin plus Ang-II in the assays? The pre-incubation should eliminate any agonist activity completely. Could the authors comment on this approach?
3. Somatostatin has been shown to modulate RAS system (https://www.nature.com/articles/292262a0) . Conopeptide agonists and antagonists of somatostatin receptors have been discovered (https://pubmed.ncbi.nlm.nih.gov/35319982/, https://pubmed.ncbi.nlm.nih.gov/23567999/). Given these observations, is it still accurate to state "There is no identified toxin targeting GPCRs involved in RAS" (line 341)?
Round 2
Reviewer 1 Report
Duplicity in Figure 6 and 8 has been fixed in a new version of the authors response.
The authors have made the appropriate revisions in response to comments made on the previous version and cleared up most of my questions. This version looks much better and I believe it is valid for publication in IJMS.